# The Life Opportunities of Young Refugees: Understanding the Role, Function and Perceptions of Local Stakeholders

Zeynep Aydar [1,2]

1   ILS Research gGmbH, Brüderweg 22-24, 44135 Dortmund, Germany; zeynep.aydar@ils-forschung.de
2   Faculty of Social Sciences, University of Duisburg-Essen, 47057 Duisburg, Germany

**Abstract:** The focus on the local level in migration research became common when analyzing arrival contexts. Despite the relative autonomy of the local level and its crucial position in the multi-level migration governance, there is limited research on the role, function and perspectives of local stakeholders in Germany. This paper investigates the dynamics of local actors and aims at understanding their contribution to the life opportunities of young refugees. A post-industrial city, namely Dortmund, has been used as a case study for this explorative task. Building on 20 expert interviews conducted between November 2020 and September 2021, the results show that while the migration history of the city has positive influence on stakeholder perspectives, there are concrete horizontal discrepancies between governmental and non-governmental actors. Albeit being engaged with inclusive migration measures for decades, the governmental actors are found to be limiting youth's chances, as they are bound to the legal framework of the national and federal levels. Contrarily, the non-governmental actors are of great importance as they challenge the system of burdens and actively create further possibilities for these youths. However, the article found that it is beyond the power of non-governmental actors to eliminate structural and legal barriers. The vertical and horizontal conflicts in multi-governance system are the major barriers for this. Nonetheless, local level actors appear critical in creating further opportunities and advocating for youth; therefore, their potential operational strength should not be undervalued.

**Keywords:** refugee integration; stakeholder perspectives; multi-level governance; local level; non-governmental actors

## 1. Introduction

Germany has been a migrant-receiving country for more than a century. In recent decades, migrant groups from many different backgrounds have arrived, driven by a range of motives. Nevertheless, Germany's history in terms of support mechanisms and integration policies is a very recent one. Describing itself as a 'country of immigration' only gained validity at the national level after 2000. Prior to that, cities, towns and even neighborhoods were responsible for designing a range of integration projects, resulting in a patchwork of in- and exclusion. Despite its longer engagement with migrant integration, research on how this local level and its stakeholders interpret, modify and in some cases contest national legal and institutional dynamics is limited. German local administrations are legally bound to the decisions and frameworks dictated by the higher administrational levels, the *Länder* (Germany's 16 federal states) and the federal government, arguably making them 'structurally weak'. Yet, municipal officials have the right to design the necessary processes at the local level, often with the support of non-governmental organizations (NGOs). This illustrates their operational strength, in contrast to their structural weakness (Schönig 2020). Against this background, this paper aims at understanding the roles, functions of local stakeholders and their perceptions of the life opportunities of young refugees. In light of the empirical results, the role of education in understanding these opportunities is highlighted in this paper, being perceived as a door opener for future prospects. Young adults (aged

18–29) are a key group when looking at multiple aspects of stakeholder involvement in a local context. They find themselves in a transformative phase in life—from school to work, from childhood to adulthood—bringing with it drastic changes in their lives. Looking specifically at refugees, their experience of flight, the interruption of their education and their arrival in a new country involve great challenges, especially when they flee without their parents. While some challenges can be overcome through peer support and social networks (Alhaddad et al. 2021), struggles with a differentiated and demanding bureaucratic and legal system in Germany, on top of them being in a fragile transformative stage in their development, put these young refugees in a vulnerable position. The local level is of particular importance for them, as it is where they receive initial support and find opportunities to socialize. From a policy perspective, their education and insertion into the labor market are prioritized in Germany (Chemin and Nagel 2020).

In line with the so-called 'local turn' (Caponio and Borkert 2010) in migration research that shifts the focus from national level to lower levels, the following sections explore the operational strength of local stakeholders in the case of Dortmund, a city in the Ruhr Region with a long history of immigration and a worthwhile location for this explorative task. This paper concentrates on refugees aged 18–29, a common target group in refugee and asylum legislation and integration work, stressing that endowing young refugees with the necessary resources is a cooperative task requiring not only multiple players, but also changes to current measures (Schütte 2016).

The following sections start by looking at the significance of multi-level governance and its relevance for understanding stakeholder structures and their roles (Section 2). This is followed by the methods and materials section (Section 3). The findings (Section 4) are presented in three parts: first, contextual and historical information on Dortmund with a particular focus on its population characteristics, migration dynamics and governance. Second, stakeholder perceptions of these contextual factors; and third the power conflicts between government and non-government organizations (NGOs) which obstruct support work. The final Section 5 presents empirical data on the operational strength of local stakeholders, highlighting their limited space of action in a multi-level system when it comes to legal matters. Regardless of such limitations, it is ultimately argued that local NGOs in particular remain important as they are able to create alternatives and further opportunities for these young refugees by circumventing with formal/legal structures.

## 2. Understanding Multi-Level Governance

The local turn in migration research has become a common focus in the literature, with its focus on cities, towns, villages or similar localities. The specific features of places are understood as important for migrants' life opportunities and thus for their integration trajectories. One main reason for this focus is that the outcomes of integration policies and support structures often depend on spatial levels lower than the national one, since specific integration instruments are adapted to the local context[1]. Further studies emphasize the local government role in determining life opportunities, integration and migration policies (Caponio and Borkert 2010), an aspect outside the focus of earlier research. Studies investigating the local level, opportunity structures and migrant integration often differentiate between spatial differences such as urban vs. rural (i.e., Jentsch 2007) or compare places with similar characteristics (i.e., Avcı 2006; Ersanilli and Koopmans 2011). Yet, looking solely at the local level is not enough to understand (life) opportunities and the legal and institutional structures surrounding them, simply because they are not the only administrative and geographical units influencing migrants. As integration policies and measures are often created and regulated at higher levels—regional and national—a multi-level approach is needed to understand migrants' life opportunities. Likewise, the multi-level governance literature is beneficial for understanding local stakeholder involvement in relation to these opportunities, as it analyzes the connection between administrative bodies, geographical levels, policymakers (i.e., local politicians) and policy users (i.e., migrant/welfare organizations). While the connections between different hierarchical levels are termed vertical, those

between stakeholders in a certain locality and/or those between different localities on the same level (e.g., cities) are termed horizontal (Matusz-Protasiewicz 2014). One key finding in this respect is that governance functions better when the relationships of administrative bodies are horizontal rather than hierarchical (Scholten 2013).

Many existing studies explore neither these multi-level connections nor the gap between local and national approaches to migrant integration. This gap, also called divergence, is explained as a gap between discourse and practice or as the failed transfer from national to local spheres (Jørgensen 2012). At the end of the day, this divergence leads to local contexts being 'richer' with regard to their abundant policy-shaping capacity and willingness. The role of local government as the policy applier and the idea of the local level as the level where life takes places are further reasons for the shift towards a more local lens—a lens increasingly supplementing or even replacing the sole focus on the nation state[2]. Municipalities, on the other hand, neither apply policies in an identical way, nor have similar connections to broader levels. The focus on the local level is therefore beneficial for comprehending these variations, i.e., distinct responses to similar problems (Zapata-Barrero et al. 2017). Consequently, a multi-level governance approach with a local focus is essential since the way the local level handles challenges revolves around the political-institutional power distribution between national and local-level stakeholders (Krummacher 2011). It is argued that the divergence and aforementioned dissimilarities have led municipalities and other organizations to take initiatives in terms of adapting national policies into more suitable versions and creating further instruments for migrants (Matusz-Protasiewicz 2014).

Despite the advantages of a multi-governance approach and the widely acknowledged need for a local turn, research following these lines varies. Dekker et al.'s (2015) overview of common multi-level governance models is key to understanding different scholarly perspectives, categorized mainly as three mainstream theories: (1) the national model, (2) the local dimension and (3) the localist model. The first attributes policymaking power solely to the nation state, viewing the local level only as the applier of national policies. In contrast, research on the local dimension supports the idea that the nation state provides policies, while allowing municipalities to apply them and learn horizontally. Finally, localist theorizations focus on specific local contexts and their uniqueness, claiming that the national paradigm does not fully apply. Developing this model further, Scholten et al. (2018) categorize two further types of governance: 'multi-level' and 'decoupling, where the former is characterized by 'vertical interactions between levels', and the latter by the 'absence of vertical and multilevel settings' (p. 2014). While it has become increasingly common to develop integration concepts and agendas for local administrations, this can either create decoupling, once divergences arise between levels, or can result in functioning multi-level governance (Scholten and Penninx 2016). Apart from these multi-level dynamics, the legal framework of the arrival context (Portes 2010) such as asylum laws (see Bosswick 2000), integration policies (see Hinger 2020) and the development of these legal aspects throughout history (see Hanesch 2016) is crucial when investigating stakeholder functions and roles. Due to space restrictions, these legal dynamics are not investigated in this paper. Instead, specific local features such as urban or rural, migration history and experience, a welcoming or hostile atmosphere, and demographics are core to exploring stakeholder engagements, their perspectives and influence on young refugees' lives (see Portes 2010; Simsek-Caglar and Glick-Schiller 2011; Dijkstra et al. 2018).

In the migration scholarship, there are multiple studies that focus on actors working in migration related sectors. Owing to the broadness of the different roles occupied by such actors, in the German literature, the focus and scope of research on these actors is also broad. Existing research has spanned from investigating the role of political actors, to the non-statuary actors, or the cooperation among them. Zick et al. (2018), for instance, identify multiple conflicts among voluntary and professional actors who are active in refugee aid. These conflicts not only consist of inter-group conflicts, but also conflicts between volunteers, institutions and the legislation. Similarly, a study on the federal state of Saxony-

Anhalt and the labor market integration of refugees specifically focuses on the cooperation between statuary and non-statuary actors and finds that a centralized knowledge of the existing actors at the regional level is missing, which causes hardships for actors to exchange and benefit from one another (Apfelbaum et al. 2020). Despite such conflicts and hardships, some research shows the potential of cooperation between migrant organizations and established institutions. Hunger and Metzger (2011), for instance, find that potential cooperation was possible as the migrant organizations had specific expertise on the target group, and the latter had access to resources. Similarly, Speth and Becker (2016) investigate the local level actors in cities of Berlin, Mannheim and Starnberg in Germany, and identify five main groups of migration actors: statuary, established organizations, civil society actors, support groups and refugees themselves. They find that the perspectives of these actors regarding one another changes positively over time, becoming more appreciative and recognizing the value of each other's work (Speth and Becker 2016, p. 9). What is more, the research report by Köhling and Stöbe-Blossey (2018) shows the uneven engagement of local actors when it comes to career orientations of young refugees, although statuary and non-statuary actors cooperate at the local level in the state of North Rhine-Westphalia. Last but not least, Adam et al. (2019) investigate the logic of local actors and highlight the space of interpretation left for localities in terms of integration policies, which creates spatial differences among municipalities. In addition, they state that national frames limit local municipalities when offering refugees various opportunities, thus, the operational logics of municipalities remain restrictive, in comparison to the civil society actors' (Adam et al. 2019, p. 349). Similarly, Schlee (2020) shows how different operational logics of statuary actors lead to conflicts regarding labor market integration of refugees. The great potential of non-governmental actors, for instance migrant organizations, is reported when it comes to supporting their target groups and filling the gaps that exist in the governance structures (Bonfert et al. 2022). Although such a line of research contributes to an understanding of the roles of certain stakeholders, and the cooperation patterns and needs among them, it does not provide a broader view on how they uniquely act in a multi-level system. Most importantly, the question of local stakeholder's impact on young refugees' lives remains unanswered.

## 3. Materials and Methods

To explore the role and structures of local stakeholders in promoting the life opportunities of young refugees in Dortmund and their structures, a two-step approach was followed. First, to understand the development of contextual factors in Dortmund, the city's background was investigated from a historical perspective. This desk research included reviewing the literature on Dortmund, its migration history, economic development, and its approach to migrant integration. To situate the local efforts and developments in the multi-level system, official reports were examined using a document analysis method (Bowen 2009). This secondary data served to prepare the ground for understanding stakeholder activities in a socio-spatial context and is introduced in the subsection of results (Section 4.1). The second step involved conducting 19 semi-structured expert interviews with various stakeholders in Dortmund between November 2020 and September 2021. The reason for choosing experts as the main sample of this article was to mirror their service provision experiences from a temporal perspective, and to hear their opinions on multi-level structures. These experts were selected through an initial stakeholder mapping process that included systematic research on existing government and non-government organizations in Dortmund offering projects, opportunities and services for young refugees (age 18–29), and their categorization by scope, focus and area of activity. This exercise identified relevant municipal officials, migrant and refugee organizations, experts (i.e., counselors and language teachers), non-profit associations, Catholic and Protestant welfare organizations, cultural centers and umbrella organizations initiating collaboration and networking. The interview sample consisted of key specialists working with migrants and young refugees and included both governmental and non-governmental stakeholders.

The composition of the institutions that the interview partners work for is diverse. They consist of four different municipal bodies (4), migrant/refugee organizations (2), non-profit associations providing youth related services (3), religious welfare organizations (2), and a university (1). In addition, two German language teachers working with various language schools participated in the interviews. It is crucial to note that, most of the employees of the interviewed non-statuary institutions are migrants or refugees themselves, and work towards similar goals. This means, if the definition by Pries (2010) on migrant organizations is considered, criteria of having goals, a structure, composition of mostly migrants, and focus on migrant-related themes- there are no concrete lines between the structures of migrant/refugee organizations and other non-profit institutions in the sample of this paper. Overall, in the institutional landscape of Dortmund, resembling of migrant organizations was to be observed, with the one exception being the religious welfare organizations, where fewer migrant employees were detected.

Hence, among the sample of the current article, 9 over 19 participants had either migration heritage, or were migrants themselves. In addition, the interviewees had diverse professions, from NGO founders to project managers, social workers as well as counselors (See Table 1). The gender ratio of the interviewees was equally balanced. On the other hand, as it can be seen in Figure 1, the proportion of governmental actors is quite small in comparison to the non-governmental stakeholders. Accordingly, the voices of NGO actors are more dominant in the empirical results, owing to such imbalance. While this can be seen as a limitation of the paper, the empirical data still provide rich information to make sense of both perspectives. What is more, among the interviewees, just one actor was in the 18–29 age bracket, reflecting the age dynamics of integration players in Dortmund and raising the question of whether young people should be included more in integration work. In terms of youth work, language acquisition, education and the labor market were identified as the three main foci for the majority of stakeholders.

**Table 1.** Professional titles of stakeholders.

| Professions of Interviewed Stakeholders | Numbers |
|---|---|
| Social Planer | 2 |
| Co-worker, Integration Center | 1 |
| Project Manager | 1 |
| Association Co-Founder | 5 |
| Co-worker, Department Manager | 2 |
| Social Worker/Counselor | 2 |
| Language teacher | 2 |
| Co-worker, Cultural Center | 1 |
| Manager/Education Consultant | 3 |
| Total number of interviewees | 19 |

Interviews were conducted mainly in German, with two exceptions in English and Turkish. They were all transcribed verbatim. Quotes were translated into English for this paper. Though using content analysis (Krippendorff 2013), the transcriptions were coded inductively in MAXQDA. The main topic focused on by stakeholders in their narratives was the concept of integration, underlining the barriers and responsibility for integration within their daily work. In describing the structure and examples of their work, a further focus was put on the historical development of opportunity structures that they, along with other institutions, provide to young migrants in Dortmund. The data were further analyzed via social environment analysis, a method originally developed by Shevky and Bell (1955) to understand the social dimensions, i.e., social space, urbanization and ethnic segregation, and indicators of urban growth, i.e., the size of the employed (female) population, fertility

rates, and numbers of foreigners, as a reflection of the increasing complexities of modern nations. Although the indicators and their generalization potential have been widely criticized in scientific scholarship, the operationalization of the method has transformed over time. The method is now applied in various disciplines, from urban sociology to social work, and not exclusively limited to quantitative data. This paper uses a research approach where social environment analysis is used qualitatively for contextual analyses of experts' behavior, opinions and value orientations (Gestring and Janßen 2002) to identify the needs of various groups, to depict inequalities and explore the development of engagement in integration work in Dortmund.

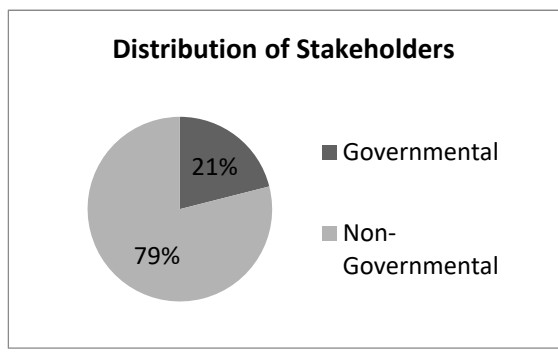

**Figure 1.** Stakeholder distribution.

## 4. Results

The following sub-sections aim to present the main findings of the conducted research for this paper. Before introducing the empirical data, the first two sections (Sections 4.1 and 4.2) serve to introduce the analysis of the literature review. This background information on Dortmund and its migration governance is followed by empirical findings on how stakeholders perceive Dortmund (Section 4.3) and the operational strength of stakeholders (Section 4.4).

### 4.1. Making Sense of Dortmund

The largest city of the Ruhr Region, Dortmund has a vibrant and diverse immigration history. From its very inception, the city's steel industry was a magnet for migrant labor, attracting migrants from the 1890s until its decline in the 1990s. The first ones came from Poland and East Prussia, while later ones came from Turkey and Morocco under *Gastarbeiter*[3] (guest-worker) agreements. Dortmund received not only migrant labor, but also other groups such as refugees from ex-Yugoslavian countries during the Balkan wars and from Syria as of 2013. As a result of these long-lasting mobility paths and economic restructuring, the city transformed immensely over the decades, developing from a city where people came to find jobs to a city where unemployment is one of the biggest concerns. 603,000 people currently live in Dortmund, 19% of whom are foreigners (defined as non-German or whose first citizenship is not German), and 17% of whom have a migration background[4], (Stadt Dortmund 2021). According to the latest detailed statistical sample in 2018, Dortmund's largest foreign populations are, respectively from Turkey, Syria, Poland, Romania and Spain. On the other hand, groups with migration background have mostly roots from countries such as Poland, Turkey, the Russian Federation and Kazakhstan (Stadt Dortmund 2019). The population dynamics and the constant influx of migrants to Dortmund are indicative not only of Dortmund's multicultural character, but also of the deep-rooted necessity to provide opportunity structures, services and integration measures for distinct groups of migrants. Owing to its migration history, the stakeholder scene in Dortmund is also diverse. The stakeholders, some of whom are themselves migrants or have a migration background, belong to various government and non-government institutions. This division is not, however, to be understood in the sense of the former being the policymaker and the latter the user, as some NGO-based stakeholders are politically active

in Dortmund, i.e., in integration councils. This overlap not only gives the stakeholders the opportunity to bring necessities on the ground to the attention of the local political level, but also makes Dortmund an interesting case study city for investigating the role and operational power of local stakeholders.

Looking specifically at Dortmund, we find many studies focused on migration and integration. Nearly all of them look at the most segregated neighborhood in the northern section of the city, Nordstadt. Now the city's dynamic and ethnically diverse district, Nordstadt was a working-class neighborhood before industrial decline set in. It experienced a drastic transformation, making it into a place marked by unemployment and poverty (Kurtenbach 2015b). Today, Nordstadt is a place full of challenges, including drug dealing and prostitution, making it Dortmund's most difficult neighborhood (Borstel 2016). A classic 'arrival neighborhood' with high percentage of immigrants, Nordstadt has been the topic of much research, e.g., on new Roma and Bulgarian arrivals to the city and their motives (Kurtenbach 2015a, 2018), on the Turkish migrant economy and entrepreneurs (Fischer-Krapohl 2013), on newly arrived immigrants' access to resources in the neighborhood (Hans and Hanhörster 2020), and on the social networks and sense of attachment of highly skilled groups (Plöger and Becker 2015). Despite an existing tradition of local-level research into migration-related themes, studies regarding Dortmund are unable to answer two key questions. The first relates to the role and engagement of stakeholders in a divergent multi-level system, looking not only at their functions, but also their interplay and their possibly uneven role in influencing the life opportunities of young migrants. Secondly, the dominant focus on Nordstadt in the literature hinders an overarching understanding of how the city and its stakeholders operate.

### 4.2. The Local Governance of Assisting Migrants

In the absence of any national integration policy in Germany, municipalities previously had responsibility for applying inclusive measures to promote integration and achieve cohesive communities. Scholten and Penninx (2016) argue that, from a sociological standpoint, the local level is expected to act upon federal and state policies and regulations, as it is the level where migrants socialize, gain employment and build their families. With no overall national integration policy available until the early 2000s, it was mainly the local level that had to face up to the challenges experienced in everyday life. Therefore, it comes as no surprise that lower spatial units assumed responsibility for equipping refugees and asylum seekers with the necessary everyday tools, from language training to healthcare, making the local level the most relevant spatial unit for them. Similarly, Dortmund's history of assisting migrants and refugees started decades ago as welfare organization-based, i.e., non-government, support, with a broad network of local stakeholders involved in voluntary service provision. Continuing after the introduction of national integration policies, their work focused on filling the gaps. At a later stage, the ongoing voluntary work was taken up by municipal measures involving the city administration, thus further widening the gap between the local and national level.

Dortmund first introduced a formalized integration concept in 2005, the 'masterplan for integration & migration'. This perceived diversity as an asset and potential, canceling out the federal state's approach of not questioning the disadvantages experienced by migrants at local level (Stadt Dortmund 2013). Its first phase (2006–2010) centered on defining integration and building networks. Between 2010 and 2012, integration became the central focus and the Integration Agency (MIA-DO, Migrations- und Integrationsagentur) was established. In 2013, MIA-DO and the Regional Office for the Promotion of the Youngsters from Migrant Families (Regionale Arbeitsstelle zur Förderung von Kindern und Jugendlichen aus Zuwandererfamilien) were merged. Four strategic fields were identified: (1) education, (2) work, (3) social balance, and (4) international city. Based on the masterplan, the city finances a range of projects, mostly run jointly with civil society stakeholders and NGOs (Stadt Dortmund 2018). Another integrative aspect of the city is that it has had a well-established Foreigner's Advisory Council (*Ausländerbeirat*) since 1972,

albeit with varying names and functions. Following the 2009 amendment of §27 of the NRW Municipal Code, municipalities were required to establish an integration council (*Integrationsrat*), leading to the initial council in Dortmund being transformed into an Integration Council (Stadt Dortmund 2021). With two-thirds of its members directly elected by the city's immigrant population, the council plays a role in determining key topics to focus on during each term of office, such as political participation, inclusion or rightwing extremism. One important initiative, as explained by one stakeholder, was the pioneering of an immigration quota system in companies providing apprenticeship schemes. The aim is to have a representative number of apprentices with migration backgrounds proportional to the city's overall migrant population. Comparing policy development in Germany and the integration work in Dortmund illustrates how divergently integration is approached. The gap between the two is decisive for understanding today's integration patterns, as well as the role of local stakeholders.

*4.3. How Stakeholders Perceive Dortmund: Diverse Narrative*

When asked to reflect on Dortmund in an arrival context, stakeholders responded with a transformative narrative, reflecting not only on the city's dynamic demographics or moments of solidarity, but also on changing (national) politics and their influence on integration possibilities. The responses point to an ongoing rearrangement of reception and the negotiations between local-level stakeholders and political structures which historically shaped the city for today's newcomers. Accordingly, Dortmund has been defined as a multicultural, open and diverse city. Its vibrant, yet entangled characteristics are best explained by one of the stakeholders:

> "Dortmund is a place, a city like America or so, we've had migration since the beginning of the 19th century, I don't know what it used to be like . . . [ . . . ] . . . Dortmund really has a multicultural society. But there are very different trends here. We also have a right [wing] scene here, which is kind of difficult. And I think this is also very dependent on poverty. We're not a rich city, but that's what makes the city so nice, in my opinion. Because it is also a pragmatic city, where, how should I say, a lot is possible, many differences can exist". (NGO co-worker)

The quotation not only mirrors Dortmund's aforementioned historical background. The metaphor of 'America' illustrates its image as an 'arrival city' (Saunders 2011) with lots of potential for newcomers. Even though the nascent rightwing scene was thematized, the quotation also refers to a harmonious co-existence. Poverty, on the other hand, indicates struggles for resources and scarce employment possibilities, possibly creating conflicts between the local populations and newly arriving refugees (Meuleman et al. 2020). The influx of refugees in 2015 is a fitting example according to many stakeholders, showing how welcoming the city is, despite the increasing populist political narrative. The scenes at the main train station were seen as a special and sentimental moment of Dortmund's recent history:

> "On arriving at the train station, the picture I saw was incredibly emotional. People gathered in the station to help the masses of refugees arriving in Dortmund. One brought some döner, another brought blankets from home." (NGO founder, Board member)

The solidarity at the main train station, followed directly by civil society self-organizing an immediate supply of aid, is telling of the city's stakeholder dynamics. While many stakeholders acted promptly to provide informal support, formalized structures appeared later on. Such a slow formal reaction is related to organizational issues, as a municipal actor reflected on Summer 2015:

> "Refugee immigration is first of all interesting; secondly it is connected with a lot of effort. That was more of a logistical issue: How do we get these people? How do we finally put them in beds? So, is it communal housing, it can be apartment, how do we get supplied? Who can take care of them? What do we do if the

communal housing situation escalates? What do we need for which group? All these was an enormous effort, but we had such great support, commitment of the normal Dortmunders anyway, but also of the other actors, who were so innovative, who really always went beyond their limits, just like the colleagues from the city [administration], that was feasible, exhausting but feasible." (Municipal Worker, Integration Center)

What is visible in this quote is that a collaboration of actors was needed for the reception of newly arrived refugees. Further, the role of local population and civil society were considered essential elements of this cooperation. On the other hand, it would be misleading to take this supportive environment for granted, as it is part of a transformative process, resulting from ongoing negotiations between politics, the local population and newcomers. Such solidarity was not, for instance, the case when the *Gastarbeiter* arrived in the 1960s. Stakeholders pointed to such historical comparisons, underlining not only perceptions of migrant reception but also the lack of services and opportunity structures:

> " . . . some Turkish immigrants who've been living in Germany for 50–60 years still can't speak advanced German. These groups were not provided with any support or opportunity structures to integrate. For those previous waves, there are still no mechanisms, nothing has changed" (NGO Founder)

The fact that these labor migrants were only supposed to stay in Germany temporarily led to thinking that inclusive migration politics were not necessary in Germany in the 1960s, reflecting the concept of deservingness (Ratzmann and Sahraoui 2021; Chauvin et al. 2013) connected to the reasons for migration and its temporality. As a result of this deficit, uneven integration outcomes are highly visible in today's Dortmund, in turn appearing as a driving force shaping the ambitions of stakeholders for supporting current newcomers. Although the statement refers to a quasi-stability in terms of established migrants which indicates an integration dilemma in today's context, the national political agenda has shifted to integrating groups who arrived post-2000. National-level legislative transformations are crucial simply because not only governance patterns but also perceptions of migration at societal level have evolved. Such developments also influence local-level practices, i.e., what stakeholders can provide and what access migrants have. The Green party's entrance into politics was a turning point, with formal structures and initial integration policies introduced in that era, triggering positive transformations. Following the 'long summer' (Kasparek and Speer 2015; Hess et al. 2017) of 2015, Germany reorganized its integration policies in 2016, starting a new chapter for refugees. Dortmund became a refugee hub for new arrivals, to be redistributed among other locations. The local level therefore clearly gained a greater role, more responsibility and space for action. Multi-level governance gained further significance through the resultant division of labor too. Dortmund's integration work is accordingly understood as a success story:

> "We have rearranged so much, changed so much, and developed so many new things. For the most part, we know how to do it [integration], I would say, clearly after 10 years you should be able to say that." (Municipal Worker, Integration Center)

As shown later, such a positive assessment of local integration work is not to be understood as meaning that all stakeholders are able to integrate all newcomers or have the capacity to do so. It refers more to the know-how gained. On the other hand, the relationship between government and non-government stakeholders in Dortmund is not free of conflicts potentially influencing the contextual framework in terms of young refugees' life opportunities. The analysis points to a malfunctioning relationship between the various stakeholders, negatively impacting their operational strength. These impaired relationships were reported as barriers faced especially by non-government stakeholders in their work, as they have limited influence on political discussions benefiting youth and especially young refugees. One key complication is the exclusion of NGOs from the multi-level integration landscape, where cooperation solely between municipal authorities and higher

levels dominates. Subsequently, the gap between local and national is constantly being reproduced, as NGO expertise is excluded:

> "In North Rhine-Westphalia, there is a dual strategy. On the one hand, there is talk about integration, 'we are the greatest, we're pouring lots of money into integration'. But only for those who are here in the municipality, everyone else is kept outside." (Social Worker, Faith-Based Welfare Organization)

This dysfunction is evident not only vertically, i.e., between the local context and the federal level, but also horizontally, i.e., between organizations within the same local context, especially when NGOs are left out of important decision-making processes (see Hoesch and Harbig 2019). This power conflict and 'being left out' not only reveal an imbalance in decision-making processes at local level, but also the exclusivist nature of the administrative bodies in the provision of services. Despite such exclusivist acts, it is crucial to acknowledge that governmental actors state the importance of non-governmental actors in supporting refugees:

> "I do a lot together with the non-statuary actors, they have another know-how, they have maybe more know-how, whereas I think so, but I am often looked at a bit sceptic. But at least they have their own know-how, because they naturally look differently at the problem areas, at the people, they look differently at the connections and they see connections that we [governmental actors] don't see." (Municipal Worker, Integration Center)

While the value of non-statuary actors is highlighted by the municipal workers, the disrupted relationship between them and the NGOs is also emphasized by them. This shows that at least some municipal workers are aware of the potential mistrust or conflicts between municipal bodies, and NGOs. Thus, a functioning collaboration of all stakeholders was idealized, benefiting the policymaking process and possibly having a positive impact on young refugees' lives. Nonetheless, a further difficulty is seen in top-down decision-making:

> "Many people up there [politicians] decide over the heads of those sitting down there [immigrants] . . . they talk about numbers . . . but not about people, about this or that individual sitting there. . . . That is very sad". (NGO co-worker, Education consultant)

> "At the local level, there are agencies with all the necessary competences, yet they're not allowed to make decisions. That's the problem" (NGO co-founder, Department Manager)

As power structures are very much linked to decision-making processes, the overall governance system was reported to be broken, especially as those stakeholders who have the know-how do not possess the necessary political power to impact governance. These vertically divergent relationships take the form of non-met needs in the everyday lives of young people.

### 4.4. The Operational Strength of Local Stakeholders Facing Structural Challenges

In the context of this horizontal and vertical disparity, investigating operational strength requires focusing on the relationship between non-government and government stakeholders, such as NGOs and municipal bodies. Despite the aforementioned exclusion from decision-making processes and the limiting governance structures, the data show that Dortmund NGOs are quite resilient, as seen for example in the case of young refugees with Duldung[5] status, who find themselves outside the statutory system in terms of service provision and entitlements. These youngsters face high uncertainty as they are not seen as remaining in Germany. They are, for instance, not entitled to integration courses, can only take up employment with the local authority's consent, and were only granted access to Ausbildung[6] (apprenticeship schemes) in 2016. Due to the inequalities faced by this group, the day-to-day work of many non-governmental actors consists of supporting them

and providing them with opportunities. The treatment of young refugees with Duldung status is a suitable case study for understanding the operational strength of NGOs. Their work focuses principally on education, as apprenticeship schemes are the only available alternative for a future prospect. These NGOs help young refugees find an apprenticeship, assisting them in searching for openings and in applying. This assistance often results in young refugees gaining an apprenticeship. However, this repeatedly ends with the refugee receiving a rejection letter from the *Ausländeramt*, the municipal authority responsible for administering foreigners, due to missing documents such as birth certificates, or to the work permit necessary for the apprenticeship not being granted. Here, NGO resilience in the face of formal structures and institutions is quite evident, as seen in the following example:

> "I went personally to the *Ausländeramt*, I really begged: why don't you at least make an exception for the young man? Duldung [status] during the apprenticeship? All they could say was 'we have over 100 young people here like him, if we make an exception, where does it start, where does it end?' Yes, but this decision has a great impact on the future of these young people ... " (NGO co-worker)

The lack of flexibility or empathy among the workers of *Ausländeramt* in such incidents was repeated by various stakeholders in our sample. Interestingly, one municipal worker reflected on how essential this institution for migrants is, and critically evaluated the employing scheme of the institution, which influences its operation negatively:

> "So many things hang in the foreigner's authority, so if you need a status clarification, then you can't get around the fact that you deal with it. But the number of staff is so little that there are often no resources for exchange ... [ ... ] Perhaps intercultural competencies are lacking, which also attracts a certain type of employees possibly." (Municipal worker, social planner)

As mentioned, young refugees were not always entitled to take up an apprenticeship. Even before the legal adjustments, local NGOs were in search of alternatives. One outstanding example of the critical role played by local stakeholders came from an interview partner, who stated that they had used their know-how to ensure young refugees gained apprenticeships. However, this again is an example of the disparity between levels, as they had to go one extra step to get the formal bodies to give the necessary permission and residency entitlement:

> "There was literally no possibility of attaining Duldung status through education. The one available possibility at that time was for us to go back with the young people to their country of origin, for example to Albania, and then re-enter the country for the purpose of taking up an apprenticeship. That's what we did. And we did it because we found the situation of the young people, of the young refugees, untenable." (NGO co-worker, Department Manager)

This quotation clearly shows how local NGO work requires extraordinary measures and expertise to create possibilities for young refugees, including travelling back with them to their country of origin. While this again exhibits resilience, the interview partner further explained how they had to ensure that these young people had sufficient financial resources to support themselves in order to gain *Ausländeramt* approval. The gap between the actual cost of living and an apprentice's wages was covered via a fundraising action initiated by the institution. The process changed after 2016, when young refugees became entitled to take up an apprenticeship. But this entailed a further financial burden, making it hard to gain *Ausländeramt* approval:

> " ... and we [NGO workers] said; 'look, there's a funding gap, we have to do something about it.' The municipality rightly said, 'of course, but this is not a municipal task, this is a federal task, it's federal legislation'. But then we said, 'But the young people here are in need.'" (NGO co-worker, Department Manager)

The urgency of needs of and the confusion over which geographical level is responsible for addressing these needs creates conflict as the example shows. Such financial difficulties and the limited capability of municipality were also raised by the municipal actors, which creates burdens for them to apply new instruments and open new possibilities for young refugees. Such difficulty leads them to apply for federal and state funds to allocate resources for standing barriers, and illustrates the standing need of multi-level cooperation between levels:

> "We often need something new. That's just the way it is, not because the existing is not good, that's okay, but if we have to equip [refugees/migrants], we can't do it from existing resources. Simply because the administration has shrunk so much in terms of personnel and because we have such a tight budget that we can hardly make big steps." (Municipal Worker, Social Planner)

It is not only the vertical governance clash which clearly illustrates the broken system. There is also a horizontal clash between NGOs and municipal institutions, as it is the latter which work with experts to detect problems and fund solutions. Yet, these very institutions limit the opportunities for young refugees on bureaucratic and legal grounds. Such ongoing and recurrent patterns which leave these youngsters with no access to any sphere arguably eventually lead to them resorting to illegal structures such as drug dealing, according to one interview partner (NGO co-worker). The operational strength of NGOs is to be understood as being only partially in a position to reduce the obstacles faced by young refugees, given the limitations enshrined in the legal framework. While they are obviously capable of creating opportunities for young refugees, the high level of bureaucracy and the legal framework in Germany hinder them in ensuring and implementing these opportunities.

There is evidently a power and capability difference between non-government and government stakeholders, with the former having less transformative power when it comes to structural and legal challenges. This discrepancy mirrors what has been argued above, that those with local expertise are disempowered in decision-making processes. On the other hand, it would be misleading to completely underestimate the relative autonomy and operational strength of Dortmund NGOs in other matters. Their longstanding experience in the field has led them to develop strategic solutions to certain challenges, if not to legal ones. This was the case when solving one particular bureaucratic barrier, namely the recognition of previous qualifications. Their recognition is key to gaining access to many sectors such as education and the labor market, and also to certain legal statuses. It is therefore of great significance for the life opportunities of young refugees. One prominent example was where a young medical doctor from Syria had to go through 18 further months of education in Germany to perform his job and pass further exams yet was still not allowed to perform his profession. His degree was not recognized due to the different curriculum followed at the Medical School in Syria:

> "When that's the case, you [local stakeholders] start searching for alternative ways to solve the problem. We managed to argue that a class lasts 45 min in Germany, but 90 in Syria, meaning that the curricula match each other in the end. In these matters, of work/residence permit, recognition of degrees, language course attendance. They [policymakers] did not do much and did not allow refugees to access these, and now we [local NGOs] try to clean up the mess, providing them with a future. The politics applied from above are constantly creating new barriers that we need to solve at local level." (NGO founder, Board member)

While NGO functional capabilities are illustrated by such examples, at the same time they clearly illustrate the decoupling between the local, state and national levels. Although both the municipal authorities and NGOs are working to integrate refugees, their different functions and dynamics revealed uneven relationships between them. The interviews revealed that the municipal players took a more bottom-up approach, creating expert networks and integrating them into neighborhood plans to be reported back to the

municipality, developing solutions and finding funds (Dortmund city official). These funds are subsequently passed on to NGOs to support projects and instruments and provide further assistance to young refugees. Yet, as shown above, this circle is dysfunctional.

The results revealed that NGOs are the ones most active on the ground, in daily contact with young refugees. They repeatedly reported that most of their work involves creating opportunities for young refugees and finding ways to overcome certain structural barriers such as access to education or red tape. These barriers derive from malfunctioning vertical and horizontal systems, namely the legal structures and local municipal institutions applying them. The divergence between local and higher levels is very much evident here, with those active at local level claiming to be 'doing their best' to overcome the inequalities created by legislation for young people, by opening up opportunities for them, providing counselling, and networking. The efforts of local NGOs point to the potential of their role in providing life opportunities to young refugees. On the other hand, the disparities are a great weakness in the overall system, a weakness that local NGOs cannot overcome alone.

## 5. Conclusions

Investigating the local level has become a dominant thread in migration research. With an increasing focus on lower levels, the scientific literature has been investigating the capabilities of local players in terms of supporting migrants. Following the localist theorizations of the multi-level governance literature, this paper investigated stakeholders' perspectives in Dortmund, looking at their role in offering life opportunities to young refugees. The aim was to understand the operational strength of local stakeholders in overcoming structural challenges. The findings reveal that the historical context of the place (Hackett 2017) and economic positioning of the city (Glick-Schiller and Caglar 2011) are significant for accomplishing the work. A city's scalar positioning with regard to immigrants is uneven due to various policies and migration histories (Glick-Schiller 2013), making it necessary to study different localities in depth. It was shown that, due to Dortmund's vibrant immigration history, service provision and stakeholder work existed before national bodies introduced any overriding instruments to support refugees and migrants. While this article drew on the multi-level governance literature and especially the clear divergence between local and national levels, it also illustrated how NGOs play a key role in supporting refugees due to their longstanding engagement. Such differentiation became necessary for understanding stakeholder dynamics in Dortmund.

As the findings show, the stakeholder perspectives present a positive picture of Dortmund in terms of what the city has to offer. The institutional settings reflect the city's diversity, as mirrored in the wide scope of youth-targeting opportunity structures offered by the various stakeholders. Similarly, the findings echo this diversity, with stakeholders underlining Dortmund's multicultural and dynamic migration history when describing the city as an arrival setting. These contextual factors were reminiscent of the mode of incorporation theory (Portes and MacLeod 1999), i.e., with refugees being extended a warm welcome in the summer of 2015 and marking Dortmund as a refugee-friendly city. Following the arguments of Penninx and Martiniello (2020) that the institutional structure and arrival context among the population are more significant than the migrants themselves in arrival processes, the findings show that the stakeholders follow a similar logic. This, of course, does not mean that this article neglects the agency of refugees, and their own engagement upon arrival.

Nevertheless, the services and support provided by stakeholders are not free of conflicts, despite the positively described characteristics and the city's longstanding engagement in creating an inclusive arrival context. This article concludes that the uneven horizontal relationships between stakeholders on the same level and their relationships with higher levels hinder the positive impact that could be achieved at local level. Such hindrance results from the vertical and horizontal positioning of the local stakeholders within their power hierarchies (Caglar 2007). Especially problematic are the conflicts between NGOs and government institutions, with the former being excluded from decision-making

processes by both municipal players and higher levels. Considering the outstanding background of Dortmund in creating horizontal cooperations with other cities and vertical relations with the state government of NRW, in developing inclusive integration programs earlier than the federal government, and in having some of its stakeholders engaged not only in local politics but also in voluntary work, such conflicts are unexpected. Consequently, Dortmund enjoys neither a complete decoupling nor full functioning multi-level governance.

As seen throughout this article, providing opportunities to young refugees and improving their future chances are very much constrained by legal structures and entitlements in Germany. Local-level stakeholders play an active role in implementing state-/national-level policies and even create further programs reflecting specific territorial needs. Due to divergence in the multi-level system, inequalities arise among young refugees. NGOs in Dortmund operate as supportive institutions to fill these gaps, leveraging their operational strength. For certain challenges, this operational strength is seen as a solution, while the legal framework is perceived as a constraint, especially when young people without a stable legal status do not have the respective entitlements, lowering their chances of integration and limiting their life opportunities (Stöbe-Blossey et al. 2019). Similarly, the findings show that NGOs are not powerful enough to overcome such structural barriers, often being hindered by municipal stakeholders bent on simply adhering to national legal frameworks. On the other hand, since it is not only through legal affairs that stakeholders can contribute to young refugees' life opportunities, this example neither implies dysfunction nor underestimates their potential. Considering NGOs' longstanding expertise and know-how regarding Dortmund and migration, their contacts with other institutions and service providers enable them to create opportunities, organize funding, and circumvent red tape. As these elements can also create a positive difference—albeit still limited—for the future of young refugees, the findings clearly point to the necessity to make a collaborative multi-level effort to achieve a shift in current policies (Schütte 2016).

While this article contributes to the literature by illustrating the different roles of various local stakeholders nested in a multi-level governance system and by ultimately providing a deeper understanding of the roles and structures of these players at local level, it also has one important limitation. While it bases its empirical investigations on young refugees, their perception of how these dysfunctional multi-level structures affect them is missing. For future research, it is of great importance to investigate the personal narratives of young refugees with a view to understanding the impact of these divergences and the role of stakeholders on their lives from their perspective.

**Funding:** This research was funded by European Union's Horizon 2020 research and innovation program (Grant agreement No. 870700). The APC was funded by the University of Duisburg-Essen. The author acknowledges the support by the Open Access Publication Fund of the University of Duisburg-Essen.

**Institutional Review Board Statement:** The H2020 Project MIMY (EMpowerment through liquid Integration of Migrant Youth in vulnerable conditions) underlying this article includes human research participants. It was approved by the Institutional Ethics Review Panel of ILS Research (29 June 2020).

**Informed Consent Statement:** Informed consent was obtained from all subjects involved in the study.

**Data Availability Statement:** To protect the privacy of our research participants, research data are not shared.

**Conflicts of Interest:** The author declares no conflict of interest.

## Notes

[1] The concept of integration is not the core of this paper but is utilized to create a lens to make sense of its empirical manifestations. This paper follows conceptualizations where integration is understood as a two-way and dynamic process, in particular those of Skrobanek and Jobst (2019) and the heuristic model of Penninx and Garcés-Mascareñas (2016).

[2] A local focus does not mean that integration dynamics are fully controlled locally (see Bommes 2018). For criticism of the 'local turn', i.e., the risks of overestimating and underestimating it, see Krummacher (2011). More reflections on the understanding of 'cities are places of integration' as a potential risk when considering personal factors, service provision and opportunity structures are equally to be found in Bommes (2012).

[3] A name given to labor migrants who came to Germany between 1955 and 1973 in search of employment under a formal worker arrangement. Since the program was initially intended to end with these migrants returning to their home countries, the name 'guest' refers to their temporary status.

[4] The concept of 'migration background' was introduced in 2005 to eliminate the sole dichotomy of 'foreigner vs. German' and to distinguish between descendants and naturalized groups, and to have policies better suited to different groups (Statistisches Bundesamt 2021). Accordingly, the term refers to a person who was not born with German nationality or has at least one parent not born with German nationality: naturalized citizens, repatriates and descendants of these groups born as German citizens (Statistisches Bundesamt 2020). In comparison to Dortmund, 13% of the overall population of Germany is foreign-born, and 27% have migration background (Statistisches Bundesamt 2019).

[5] The status of temporary suspension of deportation (*Duldung*), also called toleration, is given to asylum-seekers whose applications have been rejected, yet whose deportation is delayed due to such reasons as a lack of passport, no air connection to the country of origin, or health issues.

[6] Ausbildung is an apprenticeship scheme combining school attendance and work, where gaining theoretical knowledge is combined with practice.

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
