# Peer review of "The Life Opportunities of Young Refugees: Understanding the Role, Function and Perceptions of Local Stakeholders"

_socsci, doi:10.3390/socsci11110527_

Round 1
Reviewer 1 Report
This paper makes a valid contribution to the literature by documenting the frustrations that NGO members have while attempting to assist migrants in Dortmund.
The paper’s strength is that it succeeds in demonstrating the obstacles faced by NGOs from the perspective of NGOs. It also provides a useful description of the migratory context in Dortmund.
The weakness is that despite interviewing multiple stakeholders, most of the quotations are from NGO members. Comparatively little space is devoted to the municipal authorities or the obstacles that they face. Could they be doing more to assist migrants or are they limited by budgetary and bureaucratic constraints? It also seems to simply assume that NGOs are useful/potentially useful but it seems that the only evidence for this claim comes from the statements of NGO members.
I think that the paper should do one of the following:
1. Devote more space to the limitations of the study and reframe its contribution on the frontend. And acknowledge that it is mostly centered on the perspectives of NGOs, which is fine.
2. Keep the framing, but provide more evidence of the efficacy of NGOs and/or the perspectives of municipal authorities.
Author Response
Dear Reviewer,
Thank you for the useful comments and feedback.
Accordingly, I revised my paper. In the section where I present the empirical results, I added further quotations from municipal authorities, to enrich the empirical section, and to balance the voices of the interviewees. With such examples from the municipal actors themselves, it should be now more clear that the paper does not simply assume that NGOs are helpful/useful without providing diverse evidence. On the other hand, there is actually an imbalance among the number of municipal and NGO interviewees in my sample. Therefore, as suggested, I devoted more space to such limitation, and made this imbalance clear in the methods section.
Reviewer 2 Report
The paper aims to understand the role of stakeholders and their contribution to the lives of young refugees in the city of Dortmund in Germany.
In general, the paper is well written, the methodology applied and the work done are well exposed, the result are very interesting.
However, there are three aspects that could be improved with minor revisions.
The first concerns the background and contribution of this paper to the literature. The background concerning multi-level governance is very clear and well described in the paper, which somewhat justifies the focus of the work and underlines why this study is relevant. However, there is a lack of references to analyses similar to those of the paper, in terms of topic and actors involved. Works that have involved stakeholders in the field of migration should be mentioned in the section on background, also to underline elements of novelty of your paper compared to others.
The second aspect concerns the structure of the paper. The parts concerning the city of Dortmund and local governance (Sections 4.1 and 4.2) are well written and very interesting for framing the research. However, their placement in the results section can be misleading, since they form a kind of background to the research. I would therefore suggest either creating a separate paragraph or better specifying that they are the results of desk research and separating them from the results of the empirical work. On the part concerning the city of Dortmund, in presenting the characteristics of the foreign component of the population the authors could provide some information on the situation in Germany as a whole.
The third aspect concerns how to present some results. I would suggest the support of figures (bar graphs or pie charts) in the description of the interviewed sample, or the indication of numerical information (how many city officials were interviewed, how many project managers, etc.). The support of figures can also be helpful in the part concerning foreigners in the city of Dortmund.
Author Response
Dear Reviewer,
Thank you for the useful comments and feedback.
Accordingly, I revised my paper. In Chapter 2, you will find further references to similar studies as to my paper, which underline the distinction of my paper.
Regarding the Sections 4.1 & 4.2, I followed your suggestion of better specifying that they are the results of the desk research, and added a paragraph to the main results (chapter 4), which introduces the chapter to the reader, and emphasizes what each sub-section incudes.
In order to make a better sense of foreign-population statistics, I added the numbers from Germany, as a foot note. (5th foot note of the paper). I hope this will provide a better comprehension for the readers.
You will find some change in the methods section. Following your comments and suggestions of the information on the sample, I added further detailed information, and also some visual support (a bar-chart, and a table). With these additional visual and numeric information, I provided detailed information on the type of institutions stakeholders work for, and their professions.